

# Age, growth, and natural mortality of schoolmaster (*Lutjanus apodus*) from the southeastern United States

Jennifer C. Potts, Michael L. Burton and Amanda R. Myers

NOAA National Marine Fisheries Service, Beaufort, NC, United States

## ABSTRACT

Ages of schoolmaster ($n = 136$) from the southeastern Florida coast from 1981–2015 were determined using sectioned sagittal otoliths. Opaque zones were annular, forming March–July (peaking in May–June). Schoolmaster ranged in age from 1–42 years; the largest fish measured 505 mm total length (TL) and was 19 years old. The oldest fish measured 440 mm TL. Estimated body size relationships for schoolmaster were: $W = 9.26 \times 10^{-6} \text{ TL}^{3.11}$ ($n = 256, r^2 = 0.95$); $W = 2.13 \times 10^{-5} \text{ FL}^{2.99}$ ($n = 161, r^2 = 0.95$); $\text{TL} = 1.03 \text{ FL} + 10.36$ ($n = 143, r^2 = 0.99$); and $\text{FL} = 0.96 \text{ TL} - 8.41$ ($n = 143, r^2 = 0.99$), where $W$ = whole weight in g, FL = fork length in mm, and TL in mm. The fitted von Bertalanffy growth equation was: $L_t = 482 \left(1 - e^{-0.12(t+2.79)}\right)$ ($n = 136$). Based on published life history relationships, a point estimate of natural mortality for schoolmaster was $M = 0.10$, while age-specific estimates of $M$ ranged from 1.57–0.18 for ages 1–42.

## INTRODUCTION

The schoolmaster (*Lutjanus apodus* Walbaum 1792), a medium-sized member of the family Lutjanidae, is distributed in the western Atlantic Ocean from Bermuda to Brazil (*Mateo et al., 2010*), including the Caribbean Sea and southeast Florida, including the Florida Keys. Schoolmaster are among the most abundant lutjanids found in the waters of the Cuban shelf (*Claro & Parenti, 2001*). The species is also found in the eastern tropical Atlantic from Cote d'Ivoire to Equatorial Guinea (*Lloris & Rucabado, 1990*). The majority of juveniles settle out into seagrass beds or mangrove habitats. Adults are typically found in shallow clear waters over rocky or coral reef areas up to 60 m in depth. Schoolmaster feed mainly on fishes, shrimp, crabs, gastropods and cephalopods (*Allen, 1985*; *Rooker, 1995*) and are known to form resting aggregations during the day.

Schoolmaster are of limited to moderate importance to the southeastern United States (SEUS—North Carolina through east coast of Florida and the Florida Keys) reef fish fishery, primarily from Florida waters. Estimated total landings of schoolmaster from headboats (vessels carrying at least seven anglers engaged in recreational fishing) sampled by the Southeast Region Headboat Survey (SRHS), conducted by the National Marine Fisheries Service (NMFS) were 1,203 fish totaling 1,072 kg from 1981–2015 (K Brennan, 2015, unpublished data). The species was more commonly caught by private recreational

Corresponding author
Jennifer C. Potts,
jennifer.potts@noaa.gov

anglers from both Florida waters and the US Caribbean. Annual numbers of schoolmaster landed by anglers fishing from private recreational boats and charter boats in Florida averaged 11,167 fish from 2004–2014, and private anglers from Puerto Rico averaged 9,066 fish annually during the same time period (estimated by the NMFS Marine Recreational Information Program, MRIP, T Sminkey, 2014, unpublished data). The majority of schoolmaster landed in the SEUS are from the southeast Florida-Florida Keys-Dry Tortugas region.

Schoolmaster are currently classified as an "ecosystem species" by the South Atlantic Fishery Management Council's (SAFMC) Snapper-Grouper Fishery Management Plan (FMP) (*SAFMC, 2015*). This designation means there are no federal management regulations in place for the species, but that it is acknowledged as part of the fish species community associated with managed species of the Snapper Grouper Complex of the SAFMC . The species is managed by the Florida Fish and Wildlife Commission (FWCC) in state waters with a 10 inch (254 mm) total length size limit and a bag limit of 10 schoolmaster per person per day, within an aggregate 10 snapper per person overall bag limit, in both Atlantic and Gulf of Mexico waters of Florida. Schoolmaster are not currently scheduled for a NMFS stock assessment under the Southeast Data, Assessment and Review (SEDAR) program, likely due to low annual landings and management prioritization of other species considered more important commercially and recreationally.

There are several published studies detailing schoolmaster habitat preference (*Dorenbosch et al., 2004*; *Hammerschlag-Peyer & Layman, 2010*; *Hammerschlag-Peyer & Layman, 2012*; *Mateo et al., 2010*; *Rooker & Dennis, 1991*) and trophic and feeding ecology (*Rooker, 1995*). There is a dearth of information on basic life history such as age-growth or mortality.

Information about size-at-age and growth rates of reef fishes is valuable to fishery managers. The preferred method of aging reef fish is to use sagittal otoliths (*Manooch, 1987*). The sagittae may be read as whole structures in some species but are usually sectioned into several thin sections, which are then examined under a microscope to determine the age of the fish. Age is determined by counting alternating opaque and translucent bands deposited due to fluctuations in environmental conditions, such as water temperature. Nonlinear regression relating the measured length of the fish to the estimated age determined from the otoliths leads to the generation of growth curves, which are one of the most important inputs used in stock assessments, the procedure used by NMFS to provide scientific advice to fishery managers (K Siegfried, NMFS Beaufort Laboratory, pers. comm., 2015).

We studied schoolmaster because little is known of their life history in SEUS waters. Herein, we describe age and growth parameters and natural mortality, which are important input variables for single-species, multispecies, or ecosystem-based modeling efforts, either as species-specific data or for defining more inclusive functional groups of reef fish (*Christensen et al., 2009*). This study provides the first published information on life history parameters for schoolmaster from SEUS waters.

## MATERIALS AND METHODS

### Age determination and timing of opaque zone formation

Schoolmaster were opportunistically sampled by NMFS and state agencies' port agents sampling the recreational and commercial fisheries landings in the SEUS from 1981–2015. The age samples, sagittal otoliths, were randomly selected from all fish measured by the port agents. The otoliths were stored dry in coin envelopes. All specimens used in this study were killed as part of legal fishing operations and were already dead when sampled by the port agents, thus all research was conducted in accordance with the Animal Welfare Act (AWA) and with the US Government Principles for the Utilization and Care of Vertebrate Animals Used in Testing, Research, and Training (USGP) OSTP CFR, May 20, 1985, Vol. 50, No. 97. All specimens were captured by conventional vertical hook and line gear. Fork lengths (FL) and total lengths (TL) of specimens were recorded in millimeters (mm). Whole weight ($W$) in grams (g) and sex of the specimen from macroscopic examination was recorded for fish landed in the recreational fishery when time allowed. Fish landed commercially were eviscerated at sea, thus whole weights and information about sex were unavailable. All data recorded by the port agents for each specimen were reviewed for accuracy and provided for the study.

Otoliths were processed for age readings using the following methods. They were sectioned in the transverse plane on a low-speed saw, following the methods of *Potts & Manooch (1995)*. Three serial 0.5 mm sections were taken near the otolith core. Sections were mounted on microscope slides with thermal cement and covered with mounting medium before analysis. The sections were viewed under a dissecting microscope at 12.5X using reflected or transmitted light. The section which encompassed the core was used primarily for age reading. The other sections could be used if the core section was compromised. Each sample was assigned an opaque zone, or ring, count by two experienced readers (MLB and JCP) (*Burton, 2001*; *Burton, 2002*; *Burton, Potts & Carr, 2012*; *Potts, 1998*; *Potts & Manooch, 1999*). Sections were read with no knowledge of date of capture or fish size. We calculated between-reader indices of average percent error (APE) following the methodology of *Beamish & Fournier (1981)*.

Opaque zone periodicity was assessed using edge analysis. The edge type of the otolith was noted: 1 = opaque zone forming on the edge of the otolith section; 2 = narrow translucent zone on the edge, generally <30% of the width of the previous translucent zone; 3 = moderate translucent zone on the edge, generally 30–60% of the width of the previous translucent zone; 4 = wide translucent zone on the edge, generally >60% of the width of the previous translucent zone (*Harris et al., 2007*). Based upon edge frequency analysis, all samples were assigned a calendar age, obtained by increasing the opaque zone count by one if the fish was caught before that year's zone was formed and had an edge which was a moderate to wide translucent zone (types 3 and 4). Fish caught during the time of year of opaque zone formation with an edge type of 1 or 2 were assigned a calendar age equal to opaque zone count. All fish caught after opaque zone formation would have a calendar age equivalent to the opaque zone count.

## Growth

Estimating growth of a fish is an important step in assessing the population and determining where the species fits into the community of fishes in its habitat. The most commonly used growth model for adult fish, *Von Bertalanffy (1938)* growth model, was used to estimate the parameters fit to the observed size-at-age data. Initially, parameter estimation was done using PROC NLIN, a non-linear regression procedure using least squares estimation and the Marquardt iterative algorithm option in SAS statistical software (vers. 9.3; *SAS Institute, Inc., 1987*). Because most of the samples for this study were obtained from fishery landings, the estimate of growth at the youngest ages may be skewed due to minimum size regulations imposed on the fishery or selectivity of the fish by the fishers. For that reason, the von Bertalanffy growth parameters were estimated using a left-truncated normal probability density function on length for fish subject to the minimum size limit (254 mm TL) regulation,as developed by *McGarvey & Fowler (2002)*. For samples in this study not subject to minimum size limit, the full, untruncated nomal likelihood was used. Parameters were estimated by minimizing the negative sum of log-likelihoods with the AD Model Builder estimation software (Otter Research Ltd., Sidney, B.C., Canada).

## Body-size relationships

For weight-length relationships, we regressed $W$ on TL ($n = 256$) and FL ($n = 161$) using all available data collected by the SRHS from 1981–2015. Whole weights were recorded in grams, and lengths were recorded in mm. We examined both a non-linear fit by using nonlinear least squares estimation (*SAS Institute, Inc., 1987*) and a linearized fit of the log-transformed data, examining the residuals to determine which regression was appropriate. For length-length relationships, we regressed TL on FL and FL on TL ($n = 136$) for all aged fish in this study.

## Natural mortality

We estimated the instantaneous rate of natural mortality ($M$) using two methods:
(1) *Hewitt & Hoenig*'s *(2005)* longevity mortality relationship:

$$M = 4.22/t_{\max}$$

where $t_{\max}$ is the maximum age of the fish in the sample, and
(2) *Charnov, Gislason & Pope*'s (*2013*; hereafter referred to as Charnov) method using life history parameters:

$$M = (L_t/L_\infty)^{-1.5} \times K$$

where $L_\infty$ and $K$ are the von Bertalanffy growth equation parameters, and $L_t$ is fish length at age. The *Hewitt & Hoenig (2005)* method uses longevity to generate a single point estimate. The Charnov method, which incorporates life history information via estimated growth parameters, is based upon evidence suggesting that $M$ decreases as a power function of body size. We used the midpoint of each age (e.g., 0.5, 1.5, 2.5, etc.) to calculate age-specific $M$, because the Charnov method cannot mathematically calculate $M$ for absolute age-0.

Also, for stock assessment purposes where the integer age is used to describe the entire year of the fish's life, the mid-point gives the median value of $M$ for that age. This method is currently in use in stock assessments in the US South Atlantic (E Williams, NMFS Beaufort Laboratory, pers. comm., 2013).

## RESULTS

### Age determination and timing of opaque zone formation

A total of 140 sagittal otoliths of schoolmaster were collected fishery landings along the east coast of Florida, except one fish collected by fishery-independent sampling in Beaufort, NC. There appeared to be no bias in selection of fish for otolith removal compared to all fish sampled from the fisheries (Fig. 1). From all age samples we were able to assign an opaque zone count to 136 (97%) schoolmaster otolith sections. Four specimens were excluded because sections were illegible.

We assigned an edge type to all readable samples for our analysis of opaque zone periodicity. Schoolmaster deposited opaque zones on the otolith marginal edge March through July (Fig. 2), with peak formation in June. A transition to a narrow translucent edge occurred beginning in July. Schoolmaster otoliths were without an opaque zone on the edge from August through February, with the exception of one fish caught in December. The widest translucent edge, type 4, occurred in all months except July, which immediately follows month of peak opaque zone formation, but the rate of otoliths determined to have a type 4 edge was very low in the second half of the year compared to the first half of the year. We concluded that opaque zones on schoolmaster otoliths were annuli. Calendar ages based on edge analysis were assigned as follows: for fish caught January through July and having edge types of 3 and 4, the annuli count was increased by one; for fish caught in that same time period with edge types 1 and 2, as well as for fish caught from August to December, the calendar age was equivalent to the annuli count.

Schoolmaster otolith sections were relatively easy to interpret (Fig. 3). Agreement was good between readers. Average percent error, or APE, was 4.47% ($n = 136$), which is less than *Campana*'s (*2001*) threshold level of acceptability of 5% for species of moderate longevity and reading complexity. Direct agreement between readings was 60%, and agreement for $\pm 1$ year was 90%. An age-bias plot indicates that the second reader overestimated schoolmaster ages very minimally for ages 1–7, with an average difference between readers of 0.27 years (Fig. 4). These ages encompassed 90% of our samples (122 out of 136 fish). The second reader overestimated schoolmaster ages with respect to the first reader slightly more for older ages (average difference between readers was 0.81 years for ages 8–11). When readings differed between readers, the samples were re-examined and consensus on age readings was reached for all discrepancies. The largest discrepancy between initial readings was a difference of four for the oldest fish in our study, a 42 year old fish.

### Growth

Schoolmaster in this study ranged from 170–505 mm TL and ages 1–42, although only five fish were older than age-11 (Table 1). The resulting unweighted von Bertalanffy growth

a. Recreational fishery

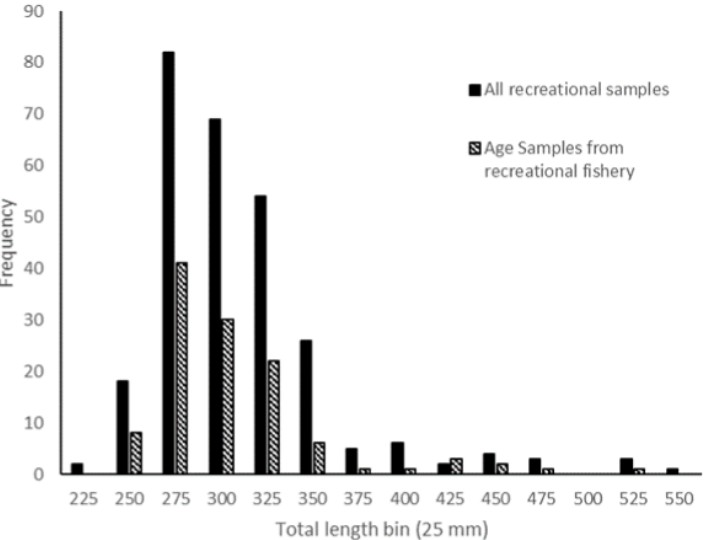

b. Commercial fishery

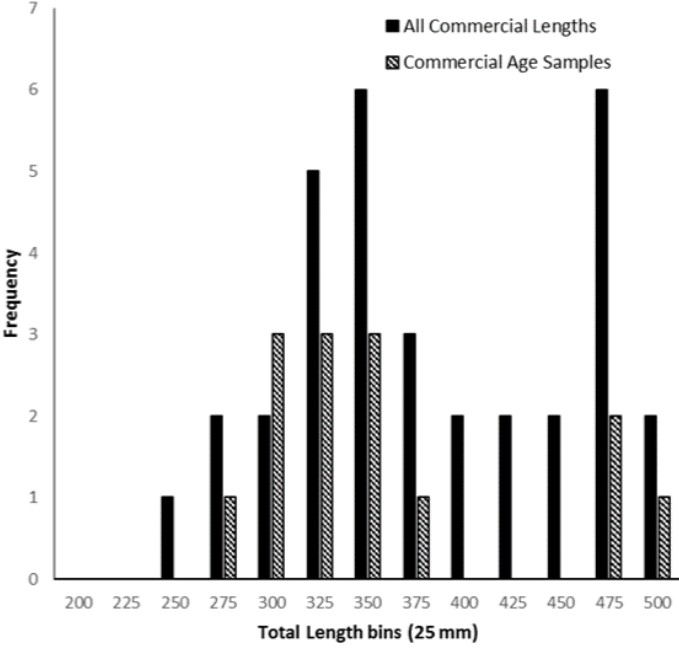

**Figure 1** **Total length frequency plots of all schoolmaster sampled from the (A) recreational and (B) commercial fisheries operating along the east coast of Florida compared to those fish selected for age structure sampling.** Total length bins (25 mm) represent the upper limit of the bin (e.g., 225 = 201–225; 250 = 226–250; etc.).

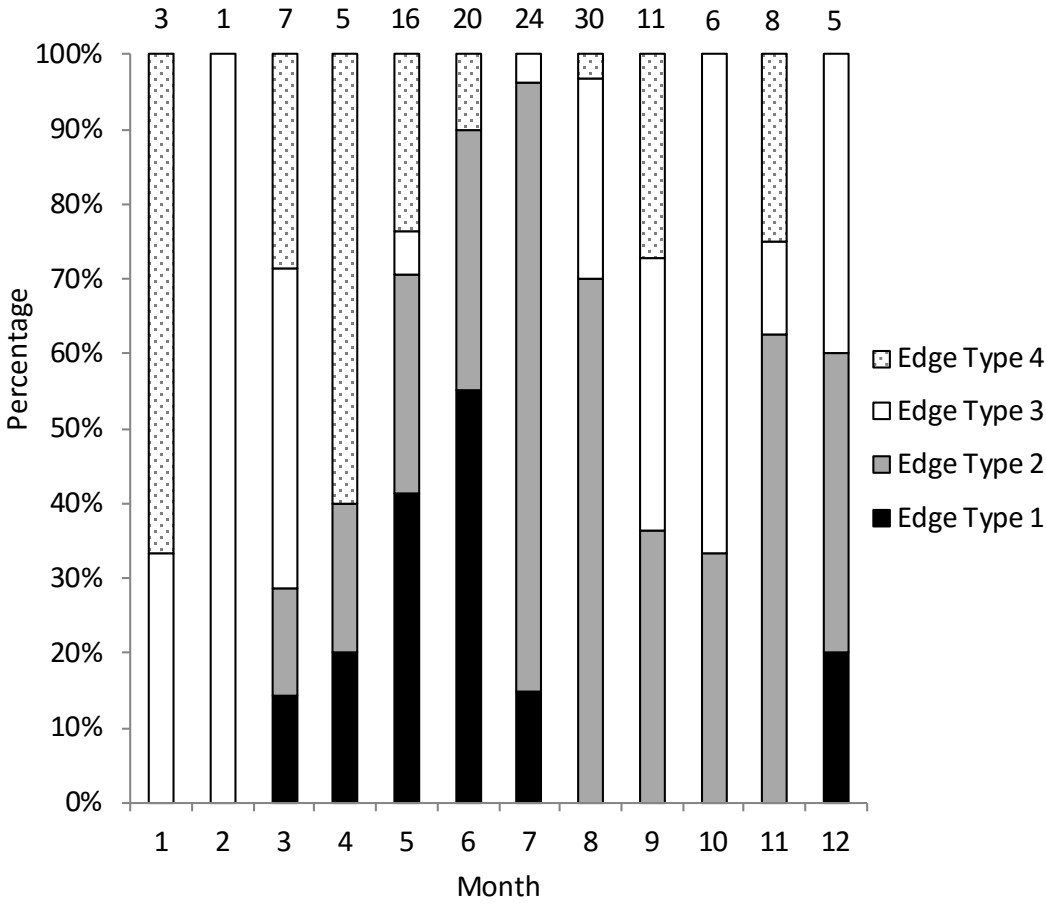

**Figure 2  Monthly percentage of edge types on schoolmaster (*Lutjanus apodus*) otoliths (*n* = 136).** Edge codes: 1, opaque zone on edge; 2, small translucent zone, <30% of previous increment; 3, moderate translucent, 30%–60% of previous increment; 4, wide translucent, >60% of previous increment. Monthly sample sizes are above each bar.

equation was:

$$L_t = 492(1 - e^{-0.09(t+5.76)}) (n = 136; \text{Fig. 5, Table 2}).$$

There was one fish less than age—2 available to us, no doubt because hook-and-line gear or fishers generally select for larger fish. Consequently, the model was unable to depict initial growth of the youngest fish, thus explaining the moderately negative estimate of $t_0$. Also, the FWCC size limit of ten inches (254 mm) TL may have excluded smaller fish from the landings and thus our samples, leading to difficulty in accurately estimating size at the youngest ages. We therefore re-ran the growth model using the method of *McGarvey & Fowler (2002)*. The resulting von Bertalanffy growth equation was:

$$L_t = 482(1 - e^{-0.12(t+2.79)}) (n = 136; \text{Fig. 5, Table 2}).$$

### Body-size relationships
Statistical analyses revealed a multiplicative error term (variance increasing with size) in the residuals of the $W$–TL and $W$–FL relationships for schoolmaster, indicating a

a.

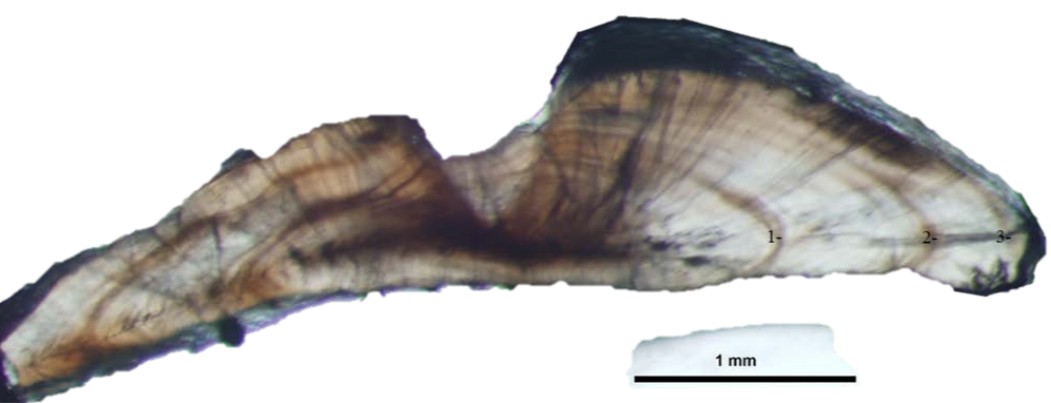

b.

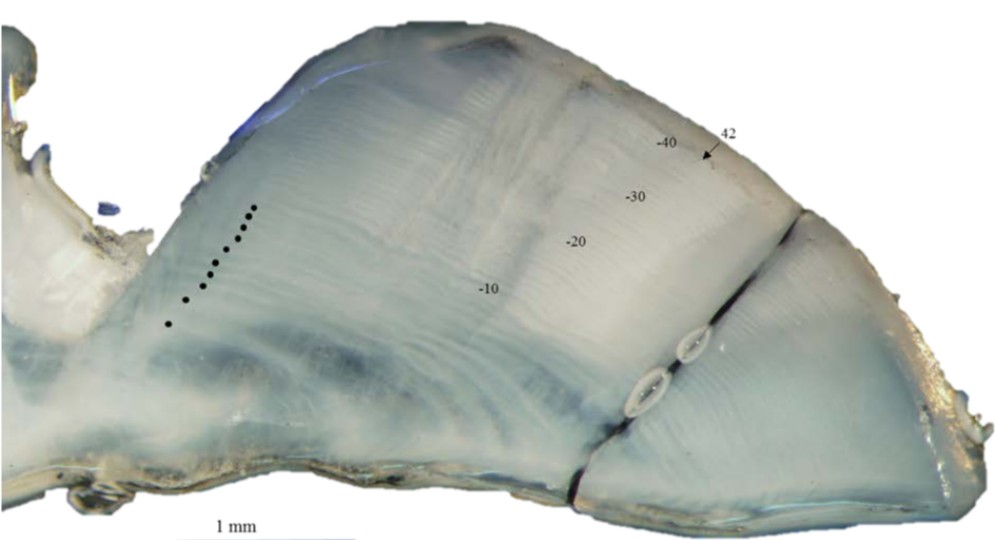

**Figure 3  Dorso-ventral section from otolith of schoolmaster (*Lutjanus apodus*): (A) 245 mm TL, age 3 yrs, and (B) 440 mm TL, age 42 yrs.** Age was determined by counting opaque increments along the dorsal axis and sulcus using transmitted and reflected light, respectively, at 12.5 X magnification.

linearized ln-transform fit of the data was appropriate. The relationships are described by the following regressions:

$$\ln(W) = 3.11 \times \ln(\text{TL}) - 11.64 (n = 256, r^2 = 0.95, \text{MSE} = 0.01)$$
$$\ln(W) = 2.99 \times \ln(\text{FL}) - 10.76 (n = 161, r^2 = 0.95, \text{MSE} = 0.01).$$

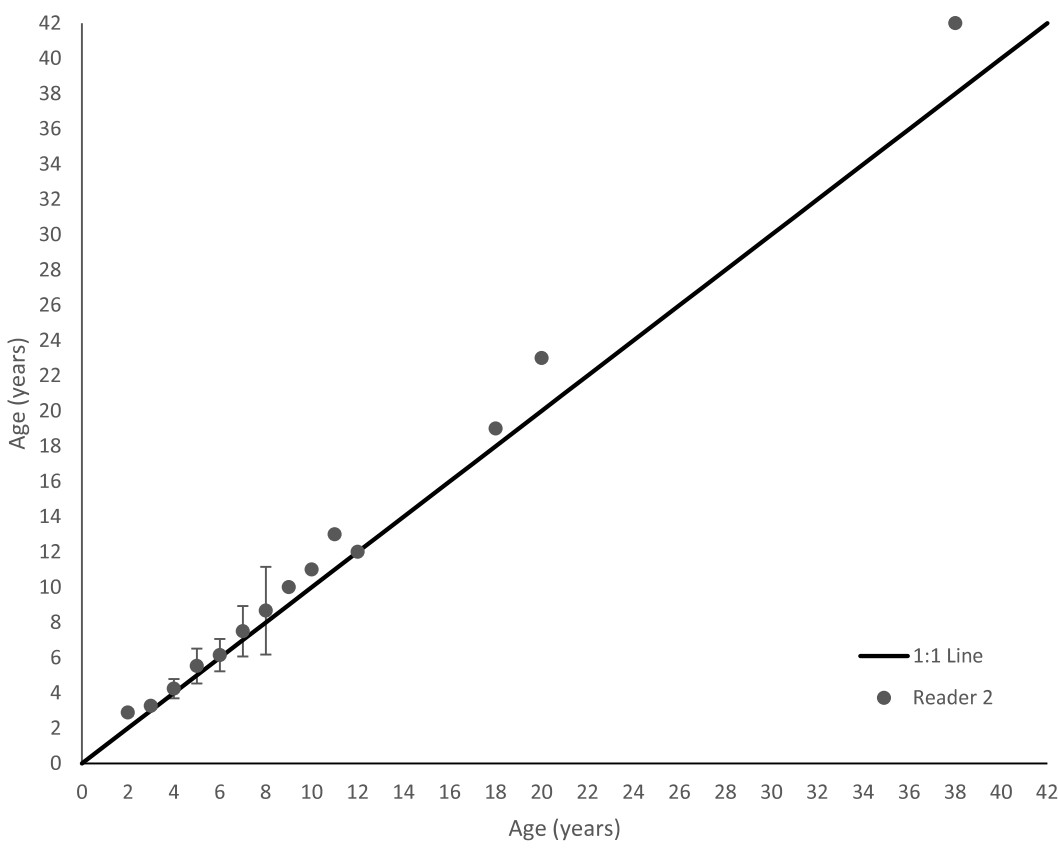

**Figure 4** **Age bias plot for 136 schoolmaster sampled from the southeastern United States from 1981–2015 and aged by two primary readers.** The second reader's mean age estimates are plotted against the first reader's age estimates. Error bars are 95% confidence intervals.

These equations were transformed back to the form $W = a(L)^b$ after adjusting the intercept for log-transformation bias with the addition of one-half of the mean square error (1/2 MSE) (*Beauchamp & Olson, 1973*), resulting in the relationships:

$W = 9.26 \times 10^{-6} \text{ TL}^{3.11}$ (Fig. 6A) and

$W = 2.13 \times 10^{-5} \text{ FL}^{2.99}$ (Fig. 6B).

The relationships between TL and FL are described by the equations:

$\text{TL} = 1.03 \times \text{FL} + 9.96 (n = 117; r^2 = 0.99)$ and

$\text{FL} = 0.95 \times \text{TL} - 5.43 (n = 117; r^2 = 0.99)$.

### Natural mortality

Natural mortality ($M$) was estimated at 0.10 using the method of *Hewitt & Hoenig (2005)*, integrating all ages into a single point estimate and using the maximum age from our study of age 42. The age-specific calculation of $M$ derived from the Charnov equation resulted in estimates ranging from 0.47–0.12 for ages 1–42 (Table 1).

**Table 1  Observed and predicted mean total length (TL) from the bias-corrected growth model, measured in millimeters, and natural mortality at age (M, *Charnov, Gislason & Pope, 2013*) data for schoolmaster (*Lutjanus apodus*) collected from 1981–2015 from the southeastern United States.**  Standard deviations of the means (StdDev) are provided in parentheses.

| Age | $n$ | Mean TL (±StdDev) | TL range | Predicted TL | $M$ |
|---|---|---|---|---|---|
| 1 | 1 | 291 | – | 185 | 0.47 |
| 2 | 5 | 249 (46) | 170–287 | 221 | 0.37 |
| 3 | 47 | 276 (26) | 245–374 | 252 | 0.31 |
| 4 | 40 | 287 (27) | 238–355 | 280 | 0.27 |
| 5 | 16 | 310 (27) | 260–360 | 304 | 0.24 |
| 6 | 9 | 322 (28) | 280–378 | 326 | 0.22 |
| 7 | 4 | 317 (18) | 300–340 | 344 | 0.20 |
| 8 | 4 | 353 (39) | 327–410 | 361 | 0.19 |
| 9 | 2 | 402 (0.73) | 402–403 | 375 | 0.18 |
| 10 | 2 | 323 (19) | 310–337 | 388 | 0.17 |
| 11 | 1 | 447 | – | 399 | 0.16 |
| 12 | 1 | 409 | – | 409 | 0.16 |
| 14 | 1 | 407 | – | 426 | 0.15 |
| 19 | 1 | 505 | – | 452 | 0.13 |
| 22 | 1 | 475 | | 462 | 0.13 |
| 42 | 1 | 440 | – | 480 | 0.12 |

**Table 2  Growth parameters for schoolmaster estimated from the freely run model and the truncated likelihood model to account for minimum size limit bias.**  Standard errors are included for each parameter.

| Method | $L_\infty$ (SE) | $k$ (SE) | $t_0$ (SE) |
|---|---|---|---|
| Freely estimated | 492 (29.20) | 0.09 (0.02) | −5.76 (1.30) |
| Size limit bias-corrected | 482 (26.43) | 0.12 (0.02) | −2.79 (0.94) |

## DISCUSSION

Otolith edge analysis demonstrated that schoolmaster deposited one annulus per year between March and July, with peak annulus formation in June. This peak is similar to timing of annulus formation for other snappers in the SEUS, which tend to form annuli in spring and summer months (*Burton, 2001*; *Burton, 2002*; *Garcia et al., 2003*; *Potts, 1998*).

The maximum age of the schoolmaster, 42 years, in this study was somewhat surprising given the small sample size available, but compared to its congeners in the same area, it was not out of bounds. Cubera snapper (*Lutjanus cyanopterus* Cuvier 1828) and red snapper (*Lutjanus campechanus* Poey 1860) have been found to live 55 and 54 years, respectively (M Burton, 2016, unpublished data; *SEDAR, 2010*). Mutton snapper (*Lujanus analis*, Cuvier, 1828) maximum age has been reported as 40 years (*SEDAR, 2015*). Gray snapper (*Lutjanus griseus*, Linnaeus, 1758) have been reported to live to 30 years (J Carroll, pers. comm., 2016). As we investigate age and growth of other lutjanids from the same waters, we may find similar longevities.
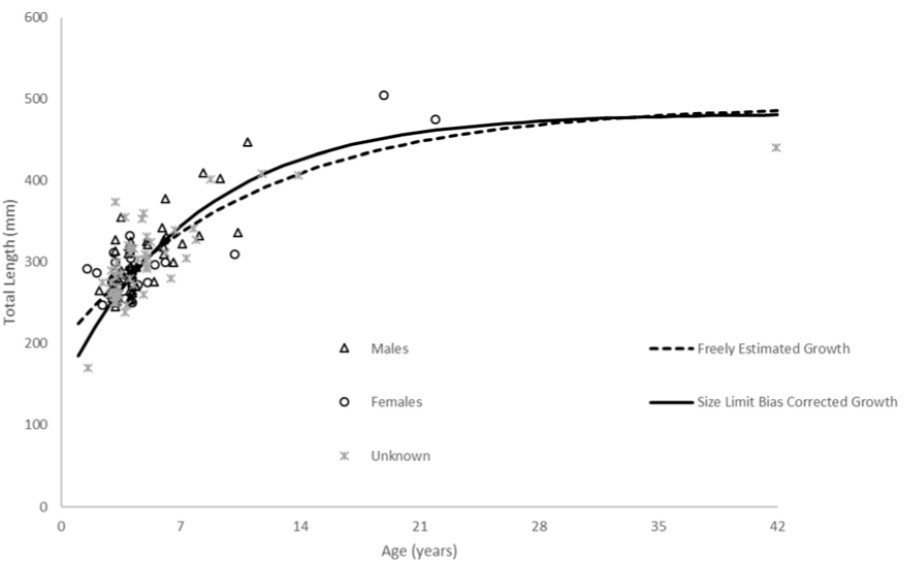

**Figure 5** Comparison of SEUS schoolmaster observed size at age to von Bertalanffy growth curves for freely estimated (unweighted) and size limit bias-corrected model runs (*McGarvey & Fowler, 2002*).

Schoolmaster grew moderately slowly, attaining an average observed size of 248 mm TL by age—2, 275 mm by age—3, and 322 mm by age—6 (Table 1). Mean observed size at age is not very informative beyond age—6 because of small sample sizes. There is moderate variability in size at age, with an average size range of 112 mm for ages 2–6 (Table 1; Fig. 5). This is not unusual for long-lived species of lutjanids. The two largest lutjanids in the US Atlantic, cubera snapper and red snapper exhibited proportionally large variability in size at age. The cubera snapper will range close to 500 mm at several ages (e.g., 470–960 mm FL range for age—8 fish, and 617–1,100 mm FL range for age—17 fish) (M Burton, 2016, unpublished data). A recent assessment of red snapper (*SEDAR, 2010*) showed similar variability in size at age (e.g., age—4, 247–865 mm TL size range; age—7, 557–905 mm TL size range). It is worth mentioning here that our oldest fish was 42 years and 440 mm, and our largest fish was 505 mm and 19 years.

Our predicted growth curve of schoolmaster using the bias-corrected growth parameters fit the observed data well (Fig. 5). The von Bertalanffy parameter $K$, or the Brody growth coefficient, which estimates the rate of attainment of maximum size, was comparable to estimates from other lutjanids from the SEUS (mutton snapper, *Lutjanus analis*—0.16 (*Burton, 2002*); gray snapper, *Lutjanus griseus*—0.13 (*Burton, 2001*)). There was little difference between parameters estimated by the unweighted, freely-estimated model and the bias-corrected model, except for the $t_0$ parameter (Fig. 5). The bias-corrected model resulted in a more biologically reasonable $t_0$ value that generated more realistic estimates of sizes at the youngest ages, thus we decided it was the most appropriate model to describe overall growth.

Some species of fish exhibit sexually dimorphic growth and should be considered when assessing the productivity of a stock. Unfortunately, the limited number of samples with the sex assigned in our data set did not allow for us to test for differences in growth. In

a.

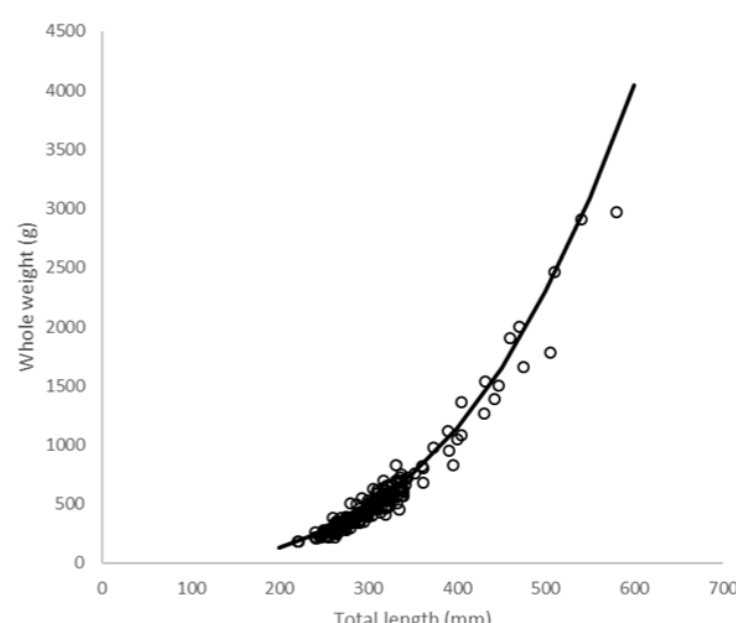

b.

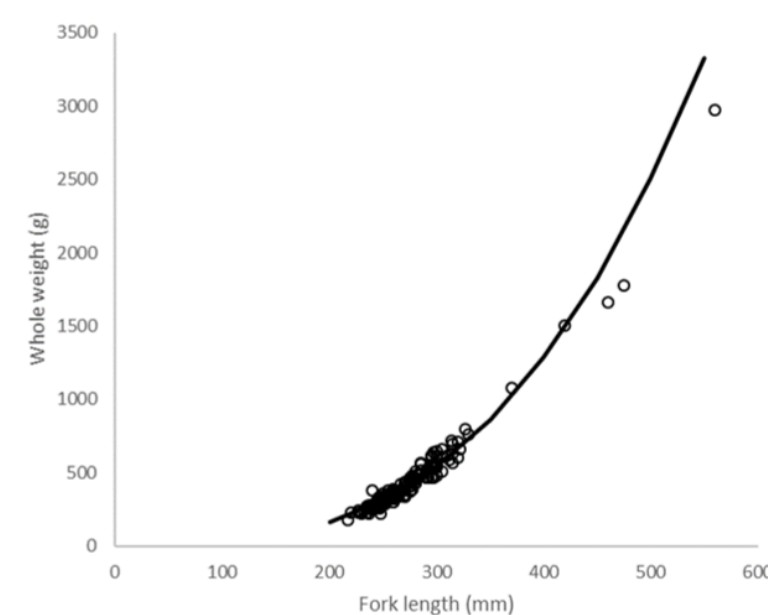

**Figure 6   Schoolmaster whole weight—length observed data and regression model fit: (A) W–TL and (B) W–FL.** Weight is in grams, and lengths are in mm.
the research conducted on the life history traits of lutjanids in the US Atlantic, generally sexually dimorphic growth has not been found (*SEDAR, 2010*; *Burton, 2001*; *Burton, 2002*). Figure 5 shows the observed size-at-age of males, females and unknown sex schoolmaster in our dataset. Even though the data are limited, there does not appear to be any evidence of dimorphic growth.

Natural mortality ($M$) of wild fish populations is difficult to measure but is an important input into stock assessments. A point estimate of $M$ (*Hewitt & Hoenig, 2005*) for the entire life span of a fish seems unreasonable, because as fish grow they become less vulnerable to predation. Given the longevity of the species, we thought that our point estimate of $M$ was reasonable for fully recruited ages in our study but was an insufficient estimate of $M$ for all ages. The age-varying $M$ calculated using Charnov seems a more appropriate estimator for the younger ages. The initial Charnov estimates of $M$ starting with the fully recruited age—4 are approximately $3\times$ the *Hewitt & Hoenig (2005)* estimate, reflecting higher natural mortality at younger ages. The age-specific estimates of $M$ for the older ages are approximately double the *Hewitt & Hoenig (2005)* estimate of $M$ by age—14 (Table 1). When considering the cumulative estimate of survivorship to the fully recruited ages, the *Hewitt & Hoenig (2005)* method estimated 2.0% survivorship, while the Charnov estimate was 0.3%. Very few of the fish in our samples were older than age—8 (9 of 143), and only two were age—22 or older. Though sample size in this study was limited, the age-frequency distribution suggests that the chance of survivorship to the oldest age may truly be lower than 1%. The two estimators of $M$ are based on different life history traits. A long-lived species would be expected to have a lower $M$, as *Hewitt & Hoenig (2005)* theorize. In contrast, given the smaller maximum size of this fish, compared to similar long-lived snappers, it may be more susceptible to predation than larger growing congeners, as *Charnov, Gislason & Pope (2013)* theorize. There is no evidence that hook and line gear is dome-selective for this species or its congeners (*SEDAR, 2010*; *SEDAR, 2015*): thus our study had the potential to collect the largest and oldest fish in the population. These observations give weight to the argument to use Charnov's estimate of $M$ at age.

One limitation of this study was the lack of fish smaller than 250 mm TL, because of the fishery-dependent nature of our samples, the selectivity of fishing gear, and the minimum size limits in place in Florida waters for schoolmaster. Lack of smaller fish is common in studies dominated by fishery-dependent samples and can lead to problems in estimating the growth curve for the youngest ages. Younger fish were unavailable to us to help define the trajectory of the growth curve at the earliest ages, so this section of the growth curve should be interpreted with caution. When the growth model was estimated with the bias-correction imposed by the *McGarvey & Fowler (2002)* method, a value of $t_0$ more representative of the theoretical size at age-0 was estimated, and the values of estimated size at age for the earlier ages were more realistic. For age—1, the estimated size-at-age was less than half of the uncorrected model's estimated size. By age—9, however, the curves intersected and were fairly similar. Our bias corrected growth curve fit the observed data moderately well (Fig. 5), given the sizeable range in length-at-age. This model resulted in a parameter estimate for $t_0 = -2.79$, which was much more realistic.

Another limitation of our study is the long period of time over which samples were collected (>30 yrs). Population parameters can vary inter-annually for various reasons (e.g., variable recruitment, environmental reasons), and it is certain that parameter estimates based on samples collected over 30 years would have increased variability when compared to estimates generated from samples collected over a much shorter time period. Unfortunately, less frequently caught species such as schoolmaster, species that are not prioritized by sampling programs because of their lack of abundance in the landings, will likely never be obtained in quantities large enough to allow us to eliminate this source of variance in the parameter estimates.

This study is the first published study of schoolmaster life history in the SEUS. We have shown that otolith sections of schoolmaster contain annuli that are relatively easy to enumerate and that otolith sections are therefore likely reliable structures for aging. Growth rings on schoolmaster sagittae are assumed to be deposited once a year in late spring–early summer, and growth is generally fast for the first seven years and then slows considerably. Our estimates of $M$ are reasonable for a fish with a moderately long life span and longevity to age 42. We believe the results of this study accurately describe the fished population of schoolmaster in the SEUS. The relatively small overall landings of this species in the commercial and recreational fisheries of the SEUS make it an unlikely candidate for a stock assessment through the NMFS SEDAR process. A more likely use of these data would be their application to studies of the population dynamics of US Caribbean stocks (US Virgin Islands and Puerto Rico). The US Caribbean is typically a data poor region and studies from the SEUS could be used as proxies in analyses for the region.

## ACKNOWLEDGEMENTS

We gratefully acknowledge the many headboat and commercial port samplers over the years whose efforts made this study possible. Nathan Bacheler, James Morris and Patti Marraro and two anonymous reviewers provided valuable reviews that greatly improved the manuscript.

### Funding
Funding for this research was provided by funds granted by the US Congress to the NOAA National Marine Fisheries Service. The funders had no role in study design, data collection and analysis, decision to publish, or preparation of the manuscript.

### Grant Disclosures
The following grant information was disclosed by the authors:
NOAA National Marine Fisheries Service.

### Competing Interests
The authors declare there are no competing interests.

## Author Contributions

- Jennifer C. Potts conceived and designed the experiments, performed the experiments, analyzed the data, contributed reagents/materials/analysis tools, wrote the paper, prepared figures and/or tables, reviewed drafts of the paper.
- Michael L. Burton conceived and designed the experiments, performed the experiments, analyzed the data, wrote the paper, prepared figures and/or tables, reviewed drafts of the paper.
- Amanda R. Myers contributed reagents/materials/analysis tools, reviewed drafts of the paper, sample processing.

## Animal Ethics

The following information was supplied relating to ethical approvals (i.e., approving body and any reference numbers):

All research was conducted in accordance with the Animal Welfare Act (AWA) and with the US Government Principles for the Utilization and Care of Vertebrate Animals Used in Testing, Research, and Training (USGP) OSTP CFR, May 20, 1985, Vol. 50, No. 97.

## Data Availability

The raw data has been supplied as Supplemental Information.

## Supplemental Information

Supplemental information for this article can be found online at http://dx.doi.org/10.7717/peerj.2543#supplemental-information.

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
