# Peer review of "Age, growth, and natural mortality of schoolmaster (Lutjanus apodus) from the southeastern United States"

_PeerJ, doi:10.7717/peerj.2543_

## Round 0.1 · original submission · Major Revisions

You have to really sort out the issues, raised by the reviewers, concerning the growth description and the growth modeling.

Reviewer 1 ·

Basic reporting

No Comments

Experimental design

No Comments

Validity of the findings

No Comments

Additional comments

Comments to “Age, growth, and natural mortality of schoolmaster (Lutjanus apodus) from the southeastern United States (#9911)”

This paper describes age and growth parameters of Lutjanus apodus from the southeastern US. Although sample size is not large, age and growth of the species is not yet studied and the information is valuable for proper stock assessment considered in the future. The oldest age 42 years is also unusual record for snappers. However, length at age data highly varied and single set of the growth parameter point estimates looks inappropriate. Authors think the variation due to the decadal change of the growth, but there are many other possibilities; e.g. 1) sexual dimorphism in growth, 2) misidentification of the species, 3) wrong measuring method for some individual fish. Authors have sex data and can consider the sexual dimorphism in growth and longevity of the species. Authors also can describe who identified and measured each individual fish. Recreational anglers may misidentify the species, and may measure fish in curved fork length or total length. Even if such problems not occurred, authors should show SD or 95% confidence intervals of VBGF parameters, it may be highly varied.

Authors should show the length frequency distribution of all measured individual fish and selected individual fish used for age determination. Sex specific data is also valuable.

Statistical support may be required to discuss annual periodicity of annulus formation (figure 1). Sample size for each month in figure 1 is also important.

Authors should show more otolith section photographs with which opaque zones are considered to be annulus. At least, the largest and the oldest fish should be shown. Relationship between otolith size and fork length is also important because age-length plots (figure 4) highly varied.

Information of natural mortality may be removed from this article because; 1) mortality parameters are just calculated by the empirical equations described by some authors, 2) current study do not used these mortality parameters further, 3) previous authors already discussed their merits and demerits (point estimates vs. age-specific estimates), 4) there are many other empirical equations. Therefore, the discussion about mortality in this paper is meaningless.

Specific comments:
L1: “Schoolmaster snapper” is FAO and Fishbase common name, should be used
L28-29: describe here how big the oldest fish and how old the largest fish
L33-35: remove the information of mortality.
L41: description style of the author of the species is wrong (see Allen 1985)
L41: the species grows up to 67 cm (see Fishbase) which size is large for Lutjanus fishes. At least, NOT “small”
L44: what does US or SEUS mean? Describe the abbreviation clearly.
L49: why not cite “Rooker 1995” here?
L52-55: what species does this sentence mention for?
L63: what does “ecosystem species” mean?
L178: no ending bracket
L179: why the result by the first author used?
L191, 204: describe the growth model and its benefit in Materials and methods
L231-234: “454 mm FL size range”? show the range like “383-451 mm”
L237: I do not think “fit well”
L250: describe “referred to as” in Materials and method
L271: why “young fish were unavailable”? Researchers collect themselves
Figure 2: show the scale bar
Figure 3-4: extend bars to 42 years
sampling area map may be added

·

Basic reporting

Unfortunately, a range of important issues exist relating to basic reporting, e.g. insufficient descriptions of the methodologies used (e.g. descriptions of growth analyses), no information regarding the precision of parameter estimates or measures of goodness of model fit, some incorrect conclusions, e.g. that growth is well estimated when there are clearly issues associated with the fit of the growth model. In the attached document, have provided some suggestions relating to how the description of growth of this species, in particular, may be improved. In summary, this is likely to require a detailed investigation to ascertain reasons as to why there are two clusters of points in the length-at-age data and use an appropriate growth model for describe trends in these data. Depending on the reason(s), this may require use of an alternative growth model along the lines described in the attached document.

Experimental design

Some issues have been raised with respect to the level of description of the methods used and reporting of results.

Validity of the findings

Given the issues raised with respect to the growth modelling, this has implications for the validity of some of the other findings reported in the paper, such as the reliability of the growth curves, and estimates of natural mortality based on life history equations based on growth parameters and other variables.

Additional comments

Please see attached review

---

## Round 0.2 · Minor Revisions

You should just follow the few remaining minor Reviewer comments.

Reviewer 1 ·

Basic reporting

No Comments

Experimental design

No Comments

Validity of the findings

No Comments

Additional comments

Manuscript is much improved.
Far more comments are follows.
Line 110: make a separated sub-section (or at least new-paragraph) for otolith preparation and ageing method
Line 170- : Results section includes some methods, should be moved to “Materials and Methods”. For examples, lines 172-177, 185-188, 236-240 and more.
Line180, 244: “in June” May is not peak
Line183: widest means type 4? 4 is also in other months
Line 289: “hereafter-” should mentioned in Methods
Line 335: lower K leads constant growth throughout their life (=more linear and growth rate not slowed)

---

## Round 0.3 · accepted · Accept

You addressed all the comments made by the Reviewer. Your manuscript is now accepted for publication.